# Blood Markers of Biological Age Evaluates Clinic Complex Medical Spa Programs

**DOI:** 10.3390/biomedicines11020625

**Published:** 2023-02-19

**Authors:** Fedor I. Isaev, Arsenii R. Sadykov, Alexey Moskalev

**Affiliations:** 1Kivach Clinic, 186202 Konchezero, Russia; 2Laboratory of Metabolomic Diagnostics of Meta-Metrix, 117630 Moscow, Russia; 3Institute of Biogerontology, Lobachevsky State University of Nizhny Novgorod, 603146 Nizhny Novgorod, Russia; 4Russian Research Clinical Center of Gerontology of the Russian National Research Medical University Named after N.I. Pirogov, 129226 Moscow, Russia

**Keywords:** medical spa treatment, biological age, aging prevention

## Abstract

Background: Kivach Clinic has developed a special medical spa program to prevent aging-related conditions in metabolic, cardio-vascular, and neurological states. Spa programs modify diet, physical activity, and lymphatic drainage, as it deteriorates with aging. We investigated its influence on the blood markers of biological age of patients during their stay to objectify the potential of spa treatment for influencing the risk of age-related events. Methods: The artificial deep learning model Aging.ai 3.0 was based on blood parameters. The change in the biological age of 43 patients was assessed after their 14-day spa treatment at Kivach Clinic. Results: Biological age decreased in 29 patients (median decrease: 8 years, mean: 8.83 years), increased in 10 patients (median increase: 3 years, mean: 5.33 years) and remained unchanged in 4 patients. Overall mean values for the entire patient group were as follows: median value was −3 years, and mean was −4.79 ± 1.2 years (*p*-value = 0.00025, *t*-test). Conclusions: The capability of specially selected medical spa treatment to reduce human biological age (assessed by Aging.AI 3.0) has been established.

## 1. Introduction

Aging is one of the most important risk factors for disease development. Modern medicine strives for healthy longevity and uses preventive measures to slow down the aging process. Biological age is the recognized method of individual assessment of aging [1]. It means an accumulation of damage by an organism, as opposed to chronological age, which is calculated by the simple passage of time. There are many ways to calculate it, including the assessment of laboratory and epigenetic markers, bone mineral density, anthropometric parameters, clinical signs, cognitive tests, etc. In recent years, computer modeling has been used, which makes it possible to use complex statistical methods for assessing a panel of biomarkers that are weakly correlated with each other and evaluating the aging of different organs, which occurs at different rates [2,3,4].

The influence of lymphatic system on the aging process is rarely evaluated. The lymphatic system maintains the drainage of interstitial fluid from virtually every cell in the body. The aging processes of the lymphatic system include the decreased contractility of the lymphatic vessels and fluid retention in the interstitial space, reduced resistance to inflammatory stimuli, and a delay in the immune response. Impaired interstitial fluid dynamics prevents the delivery of nutrients to the cell and contributes to the accumulation of toxic metabolic products and their release into the surrounding tissues, as well as to impaired pathogen transport up to their retrograde delivery through the lymph [5]. This accelerates the oxidative stress, inflammation, and aging, completing a vicious cycle.

Professor Yu. M. Levin (Department of Clinical Lymphology and Endoecology, Peoples’ Friendship University of Russia, Moscow) developed the concept of medical spa program to restore the lymphatic system function. The concept uses medical and spa treatments to stimulate lymph excretion by the lymphatic capillaries of the intestinal villi along with intestinal sorption to eliminate reabsorption and cleanse the intestines. It also affects other organs of excretion and detoxification, namely liver, kidneys, skin, and lungs. This method is used as a complex medical spa program at the Kivach Clinic. In this study, we intend to evaluate the impact of special spa treatment on biological age.

To calculate the biological age, sets of markers are used that reflect the physiological and functional parameters of the body that change during aging (significantly correlated with chronological age). For these calculations, the methods of mathematical modeling—the multiple linear regression, the principal component analysis, the Hochschild’s method, and the Klemera and Doubal’s method—were initially used [3,4]. In recent years, deep learning methods represented by neural network ensembles have become widely used to calculate biological age based on blood parameters, corner of the eye images, the Gaussian regression model based on MRI scans of the nervous system, etc. [6]. For this study, biological age was estimated using Aging.AI (3.0 version of algorithm), deep learning-based blood aging clock [7,8].

## 2. Materials and Methods

This was a single-center prospective open observational study conducted at Kivach Clinic CJSC (village of Konchezero, Republic of Karelia, Russia) between 1 February 2022 and 12 June 2022. It included 43 male and female subjects aged 24 to 61 years (median age 50 years), who underwent 14-day spa treatment under the Detox program. Four of all the patients indicated they are smokers to their attending doctors.

Patients received identical treatment: calorie restriction diet, gallbladder cleanse (tubage), cleansing enemas and hydrocolonotherapy, enterosorption, taking herbal lymphatic drainage and antiparasitic drugs, as well as microbiota restoration drugs, hydrotherapy, halotherapy, massage, systemic magnetotherapy, thermal procedures, and body wraps.

The program consists of medical and spa procedures according to a standard protocol. Medical procedures are: 3 stimulation of biliary excretion procedures (tubages), 3 instrumental colon cleansing procedures with 7 enemas, 12 therapeutic back massages, course of 5 cryotherapy sessions and 7 systemic magnetotherapy sessions. Spa procedures are wrapping (4 wet and 2 salt wraps) 12 hydrotherapy procedures (jacuzzi, bathtubs, showers at patient’s choice), thermal procedures (2 Finnish saunas with body peeling and daily available infrared sauna), 13 fitness sessions. All mentioned drugs are orally given herbal food supplements or teas except gastrointestinal adsorbent Enterosgel registered as a medication. All the information about their composition and courses are given in Table 1. All patients were examined by Clinic’s physicians to exclude possible contraindications for such therapy. All patients were observed daily by a physician during the treatment.

All patients underwent a low-calorie gluten-free diet corresponding with the food combining diet principles under observation of a dietologist. Food for the first three days consisted of vegetable and fruit decoctions (200–250 kcal per day). Then, the diet was gradually expanded with mild food (puree soups, fruit mousses, vegetable salads) from 400 up to 1200 kcal per day for the next four days. The second week diet brings proteins back to the ration such as eggs, fish, and poultry meat with 1600–1700 kcal per day restriction. Biological age was calculated using the blood aging clock Aging.AI 3.0 [7,8] by 19 parameters: albumin, glucose, urea, total cholesterol, total protein, sodium, creatinine, hemoglobin, total bilirubin, triglycerides, high-density lipoproteins (HDL), low-density lipoproteins (LDL), calcium, potassium, hematocrit, MCHC, MCV, platelets, and red blood cells. Population, height, weight, and current smoking were also used as input parameters. The mean absolute error of the model is 5.9 years [7,8]. Venous blood served as a material for the evaluation. It was taken in the clinic’s procedure rooms in properly color-coded vacuum blood collection tubes and prepared in the laboratory in compliance with national standards. All parameters were measured Sysmex KX-11 and Sapphire 400 Premium platforms at licensed in-house laboratory. The laboratory undergoes an audit of Federal system of external assessment of the quality of clinical laboratory research for every 3 months. Internal laboratory research’s quality audit carried out daily according with Ministry of Health’s orders and National Standart GOST 53133-2008.

For calculating biological age, data from the medical information system database were entered into an electronic form. The data were entered manually by two clinic employees working in different localities, released from their job duties for the period of data processing and not in contact with each other. Reconciliation of the biological age calculation results was performed by a third party. In 2 out of 86 cases, results were found to be inconsistent, the figures were recalculated and reconciled by all three employees.

The calculation of average values was carried out in Microsoft Excel 2019 using built-in tools (MEDIAN, AVERAGE, CORREL functions with selection of the corresponding data ranges for calculating the median mean, mean and Pearson’s linear correlation coefficient, respectively).

Principal component analysis for implied percentiles was produced using the R programming language, version 3.6.1, in the RStudio programming environment, version 1.2.1335. The ggplot2, ggfortify, and lfda libraries were used for the analysis. All results were assigned an implied percentile in the population. Its calculation assumed that the distribution of each of the analytes was lognormal. The reference ranges were taken from INVITRO laboratory and calculated in accordance with GOST R 53022.3-2008, that is, they were the values of 2.5 and 97.5 percentiles.

To assess the statistical significance of changes, we applied *t*-test for 2 dependent means, two-tailed hypothesis.

The study was approved by the Ethics Committee of Kivach Clinic CJSC: “The Ethics Committee approves the conduction of the study” (Study Approval Decision No. 2 dated 25 February 2022).

## 3. Results

In the group, 67.4% of responders were identified whose biological age was decreased with a median decrease of 8 years from the baseline. Out of 32.6% of non-responders, 9.3% had no change in their biological age and in 23.3% it increased with a median increase of 3 years. In 80% of cases, the increased biological age did not exceed the chronological age. The mean decrease in biological age for the entire patient group was as follows: median was −3 years, and mean was −4.79 years ± 1.2 years (*p*-value = 0.00025, *t* is −4.004217, *t*-test).

Biological age changes are presented in Figure 1 considering real patient’s age. All data are given in Appendix A.

Weight and body mass index of patients were analyzed. Patients’ weight at arrival was in the range of 54–150 kg, mean value is 83 kg. The body mass index (BMI) was in the range 20.02–38.63 and mean value is 27.13. Patients’ weight at departure was in range of 51–145 kg, mean value is 80 kg. The body mass index (BMI) was in the range 19.2–37.23 and mean value is 26.86. Nobody increased his or her weight. Only one patient had a BMI 19.2 considered as a weight deficiency at departure. The group of those who decrease biological age has a mean BMI value 28.68 (in range of 20.08–28.63) and mean value of weight drop is 4 kg. The group of patients who increase biological age has a mean BMI value 23.74 (in the range of 20.02–29.07) and mean value of weight drop is 2 kg. Pearson’s linear correlation coefficient between weight (BMI) and changes of biological age is −0.47 and −0.5, respectively.

Using Fisher’s linear discriminant analysis, two classes (patients before and after treatment) were divided according to the principal component analysis, as shown in Figure 2.

This PCA biplot (Figure 2) is a combination of loading plot and PCA plot. Nineteen vectors correspond with each laboratory parameter’s contribution to both principal components. The values of their projections to every principal component show parameter’s impact on it. The angle between the vectors reveals the level of correlation between the parameters: 0 degree means positive correlation, 90 degrees means zero correlation and 180 degrees is negative correlation. Every plot point distinguishes a patient’s laboratory profile (red points for arrival profiles and green for the departure ones). The biplot proves the second principal component (PC2) mainly contributes to the distribution into arrival and departure groups. Table 2 shows a numeric description of the loading plot (Figure 2) and the strength of contribution for each parameter in the principal component calculation. It means that total cholesterol, low-density lipoproteins, glucose and triglycerides, which make the greatest contribution to the change in biological age, correlate most of all with the second principal component as shown in Table 2.

Local discriminant Fisher analysis is a statistical method that aims to distinguish two or more object classes better. The results are presented in Figure 3. 

Patient’s laboratory profiles at arrival and departure are such groups. Every plot point is one lab profile (red points are arrival profiles and green are departure profiles). Color area is a zone where a blood markers profile of corresponding group is located with the maximal probability. We see the areas slightly overlap each other assuming significant difference of the profiles between two groups.

Responders and non-responders were additionally compared by the difference in analyte deviations, as shown in Figure 4.

Spearman’s correlation matrix displays correlation coefficients between lab analytes inside a profile and biological age. The circle size indicates absolute values of correlation coefficients. Colors show negative for orange or positive for blue direction of a correlation. We found biological age correlates with cholesterol, HDL, and glucose mostly. They are the same analytes what brings the most contribution to primary component 2 formation.

## 4. Discussion

We analyzed the influence of the Kivach Clinic’s 14-day spa programs on changes in biological age in 43 patients. Anthropometric and hematological data were used to calculate biological age using the deep learning-based blood aging clock Aging.ai 3.0. It decreased, remained unchanged, and increased in 29, 4, and 10 cases, respectively. The decrease in age was more significant (with a median of 8 years) than its increase with a median of 3 years, which leads to an average decrease in biological age in the group with a median of 3 years.

All patients were divided into groups according to their gender and age in compliance with WHO classification [9]. The results of the analysis are presented in Table 3.

Comparing two major groups of age 25–44 and 45–60, we observed an almost equal proportion of those who decreased their biological age (71% and 68%), but it is 2–3 times more efficient in 45–60 age group. In those who increased their biological age, we observed a two times more significant rise in biological age in the 45–60 age group also. All patients who increased the values of their biological age were women. Because of all the women should visit a gynecologist in the Clinic (and 20 of 23 did it) their gynecological anamnesis was analyzed. None of the sex-specific trends (menstrual cycle phase, menopause, hormone therapy) was found to be connected with changes. 

These results are consistent with information on biological age decrease associated with diet, adherence to healthy eating patterns, and/or exercise [10,11]. Stimulation of lymphatic drainage suggests improved age-related parameters such as inflammation, cognitive decline, the level of cholesterol deposition in the wall of blood vessels, and the development of cardiovascular diseases, which also contributes to a decrease in biological age [12]. Since diet directly affects gut microbiota and intestinal permeability, a change in biological age can also be due to a change in the microbiome composition [13] along with the direct effect of cleansing procedures and enterosorption to prevent the lymph and bile reabsorption. These mechanisms are the basis of the concept of spa treatment programs at Kivach Clinic according to the method of Yu. M. Levin. 

We found a weak linear correlation between the chronological age of patients or the initial difference in chronological and biological age with subsequent changes in results (0.33 and −0.41, respectively). This suggests an age-nonspecific effect of these complex medical spa programs without a prognostic relationship between changes in biological age and the initial rate of aging. 

The linear correlation between weight/BMI changes and biological age changes is near −0.5 proving some connection of weight loss efficiency with metabolism improvement and changes of aging processes.

In this study, all patients who showed an increase in biological age were females aged 33 to 61 years. When clarifying the gynecological history, there was no relationship between the dynamics of biological age and the phase of the menstrual cycle.

In 80% of cases, the increase in biological age occurred in patients whose biological age was initially less than the chronological one. No other factors were identified that could be associated with an increase in biological age (chronological age, anthropometric data, initial ratio of any laboratory markers, reproductive status, smoking). A possible reason is the individual perception of the stressful impact of calorie restriction, deviation from the usual daily routine, and the fact of undergoing medical manipulations. There is evidence that stress contributes to the acceleration of biological age, but the nature of stress-induced changes is extremely individual and depends on personality factors [14,15]. Cortisol can influence the cells of the immune system, carbohydrate and protein metabolism, the level of inflammation in the body, that is, almost all measured laboratory parameters [16]. Thus, one of the hypotheses explains this as a stress-mediated individual variability in the body responses. In this case, we assume a short-term transient nature of the increase in biological age.

Possible scenarios also include a different degree of efficiency of lymphatic drainage and a heterogeneous level of accumulation of toxic metabolites in the body. Stimulation of their release from the intercellular space into the blood can enhance the processes of oxidative stress, which increases all aspects of the aging process, and provoke an associated transient jump in biological age [17]. This study was not designed to reassess biological age after departure.

When comparing responders and non-responders (Figure 2), differences were revealed in the group of analytes associated with water-salt metabolism (urea, sodium, potassium, hematocrit). Dehydration of cells and tissues is associated with cell senescence and aging of the body [18,19]. Thus, the key and manageable difference from non-responders may be insufficient water drinking, since the water drinking schedule was not monitored in this study. This can cause the overestimation of biological age observed in this group. 

The error rate for the model (5.9 years) is not considered as a limitation for the study. The model error evaluates mean deviation of estimated biological age from real chronological age. However, it does not determine a model, or its results are incorrect because biological age of persons cannot equal their chronological age and should not be close to it neither. This research evaluated only biological age difference between two calculations of it—not the difference between chronological and biological age involving the model error. Two groups of evaluations at arrival and at departure have statistically significant difference of *p* values. Additionally, the distribution of results is not chaotic. The trend for more significant biological age changes in older patient groups is observed and only women increase biological age values. So, we conclude the programs can reduce human biological age.

## 5. Conclusions

The capability of 2-week complex medical spa programs, including calorie restriction diet, medical spa (physiotherapy for gallbladder cleanse and enterosorption, stimulation of the excretory organs), herbal medicine, hydrotherapy, and thermal procedures, to reduce human biological age (Aging.AI 3.0) has been confirmed. Medical spa programs demonstrate a possibility to improve the quality of aging.

## Figures and Tables

**Figure 1 biomedicines-11-00625-f001:**
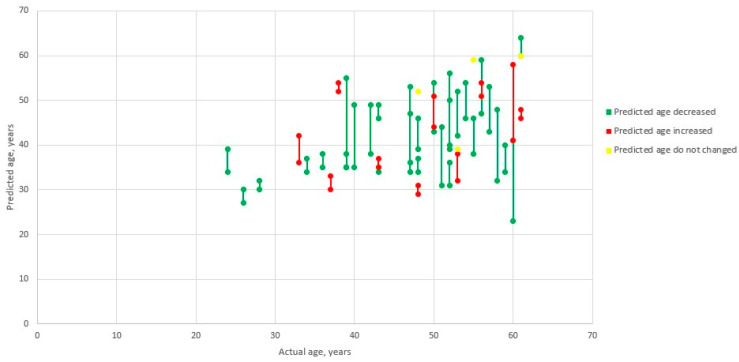
Predicted biological age data rows.

**Figure 2 biomedicines-11-00625-f002:**
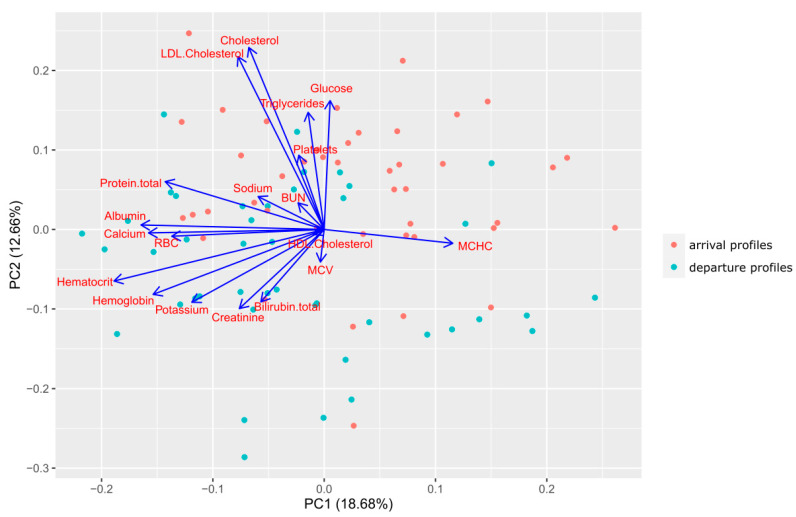
Primary Statistical Blood Biomarkers Analysis with Principal Component Analysis.

**Figure 3 biomedicines-11-00625-f003:**
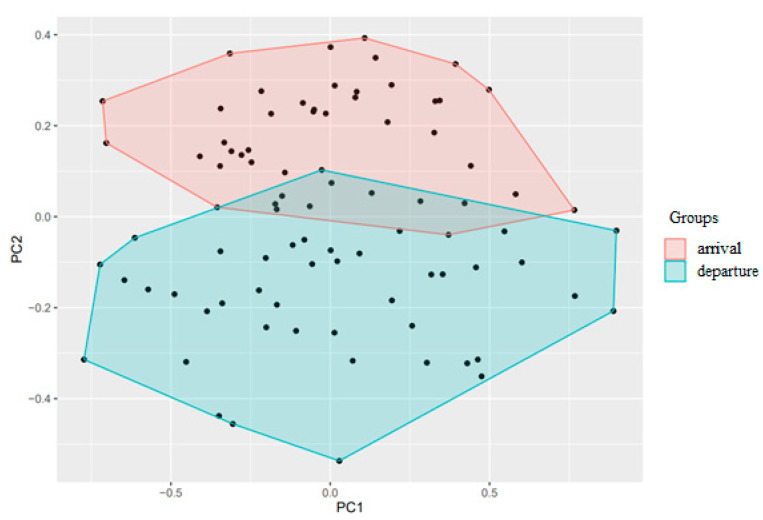
Patient Groups Analyzed by Principal Component Analysis with Local Discriminant Fisher Analysis.

**Figure 4 biomedicines-11-00625-f004:**
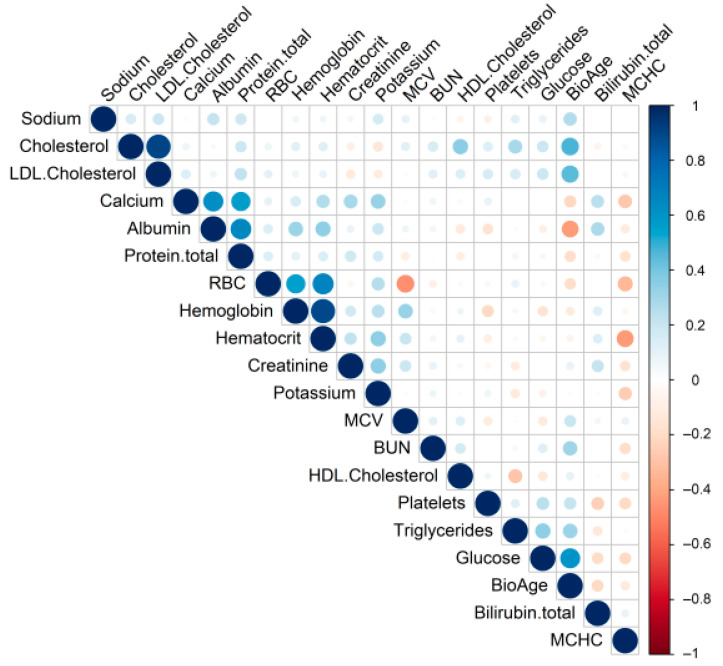
Correlation matrix of pair Spearman’s coefficients for biological age measurement blood markers.

**Table 1 biomedicines-11-00625-t001:** Drug Therapy Prescribed in this Study.

Drug Name	Type	Active Ingredients	Dosage and Course
Altai	Herbal tea	Broad-leaved Sage leaves, Common lungwort herbs, Thyme herbs, Common Plantain leaves, Scotch Pine buds, Peppermint leaves, Common licorice root, Common St. John’s wort herbs	125 mL 2 times per day (5 days)
Hepalimfol	Food supplement for herbal lymphatic drainage	Black Currant leaf extract, Silymarin (Milk Thistle seeds extract)	1 caps. 2 times per day (1 month)
Ortosifon	Herbal tea	Orthosiphon Staminate leaves	125 mL 2 times per day (5 days)
Vermitox	Antiparasitic food supplement	Tansy flowers, Carnation flowers, Walnut leaf extract, Ginger root, Centaury extract, Peppermint leaves, Dropwort extract, Birch leaf extract	1 caps. 2 times per day (1 month)
Enterosgel	Medication; enterosorbent	Polymethylsiloxane polyhydrate (methylsilicic acid hydrogel), purified water	3 tablespoons (dosage can vary because of patient’s weight) 3 times per day (4 days)
Actoflor-S	Food supplement; metabiotic for restoration of intestinal microflora	L-aspartic acid, Glycine—2.4 mg; L-leucine; L-alanine; L-methionine; L-valine; L-glutamic acid; L-lysine hydrochloride; Formic acid; Succinic acid; D, L-lactic acid	1 dropper tube 2 times per day (1 month)
Liquid Chlorophyll	Detoxication food supplement	Sodium Copper Chlorophyllin (Chlorophyll)	1 teaspoon 1 time per day (13 days)
Rhodiola	Stress-preventing food supplement	Rhodiola Rosea extract	1 caps. 2 times per day (1 month)
Kivach D3 + K2	Vitamin D3 and K2 food supplement	Vitamin K2 (Mena Q7 PharmaPure 1500 ppm MCT oil), vitamin D3 (Cholecalciferol)	Dosage is established by a physician after blood test personally according to the official guidelines

**Table 2 biomedicines-11-00625-t002:** Spearman’s Correlation Coefficients of Blood Biomarkers for Calculation by Principal Component Analysis.

Indicator	PC1	PC2
Albumin	**−0.36566**	0.012592
Glucose	0.011897	**0.360175**
BUN	−0.05212	0.074377
Cholesterol	−0.15044	**0.508855**
Total protein	**−0.31748**	0.133822
Sodium	**−0.13193**	0.091889
Creatinine	−0.1694	−0.22119
Hemoglobin	**−0.34155**	−0.18148
Total bilirubin	−0.1264	−0.20244
Triglycerides	−0.03203	**0.32749**
HDL cholesterol	−0.0222	−0.01384
LDL cholesterol	−0.17211	**0.483254**
Calcium	**−0.35063**	−0.00962
Potassium	−0.26431	−0.20353
Hematocrit	**−0.41967**	−0.14412
MCHC	**0.256823**	−0.03866
MCV	−0.00752	−0.09099
Platelets	−0.05066	0.207514
RBC	**−0.30433**	−0.01897

Correlations with *p*-value < 0.01 shown in Bold.

**Table 3 biomedicines-11-00625-t003:** Gender Group Analysis.

	Under 25 Years	25–44 Years	44–60 Years	Higher 60 Years
Decrease (Male)		6 (−3; −7.2)	11 (−11; −11.4)	1
Decrease (Female)	1	4 (−2.5; −5)	6 (−10; −9.8)	
Increase (Female)		4 (2.5; 3.25)	5 (6; 7)	1
Do not change (Female)			1	1
Do not change (Male)			2	

Median and mean biological age changes are given accordingly for each major group in brackets.

## Data Availability

All data is available as Appendix A.

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
