# Peer review of "Blood Markers of Biological Age Evaluates Clinic Complex Medical Spa Programs"

_biomedicines, 2023, doi:10.3390/biomedicines11020625_

Round 1

Reviewer 1 Report

Isaev. et al describe the use of biological age calculation from artificial deep learning to evaluate the efficacy of medical spa treatment on biological age as opposed to chronological age. Subjects underwent an extensive 14 day treatment program and 19 different parameters were quantified for use in determining biological age using the Aging.AI 3.0 clock. The authors report a modest change in biological age, however no control group was present to compare to. Due to this and other comments that need to be addressed (see below), the claims made in this paper need more validation and thus the manuscript should be rejected.

Comments

-This study lacks a control arm (to show random changes over a 14 day period) and needs further explanation of experimental design.

-Please include the description of ethical approval of the study in the methods section.

Please provide more detail on how the 19 parameters were measured (platforms, in-house vs reference labs, etc). 

-Were there changes in weight/BMI after treatment?

-It is very important to plot the changes for subjects for each parameter and use of simple statistics. It is important to see these parameters and the statistical changes within and not just purely rely upon principal component analysis.

-Please provide more specifics on drugs (route, source, dose, frequency, etc.) that the subjects were given.

-Analysis of sex-specific effects should also be presented.

-How do the biological age changes from this intervention compare to age changes from other interventions. 

-The author should reiterate the limitations of the study (no control group, the error rate for the model (5.9 years)) in the discussion section.

-“The capability of 2-week complex medical spa programs, including calorie restriction diet, medical spa (physiotherapy for gallbladder cleanse and enterosorption, stimulation of the excretory organs), herbal medicine, hydrotherapy, and thermal procedures, to reduce human biological age (Aging.AI 3.0) has been confirmed.” 

The mean error rate of the model was nearly 6 years. So it is a bit hyperbolic to say biological age was reduced.

Author Response

Point 1: This study lacks a control arm (to show random changes over a 14 day period) and needs further explanation of experimental design.

Response 1: We agree a contorl arm is important and study lacks it. Unfortunately, the patients on other Clinic’s programs or patients directed to the hospitals because of exacerbation of the disiases could not be used as a relevant contorl group. The literature observation was performed and a relevant control group of simular age and gender without significant diseases was not found to compare the results. We hope this study can inspire other reseachers to investigate methods for improving quality of people’s aging and accumulate such data.

Point 2: Please include the description of ethical approval of the study in the methods section.

Response 2: The citation was included into the methods section: “The study was approved by the Ethics Committee of Kivach Clinic CJSC: “The Ethics Committee approves the conduction of the study” (Study Approval Decision No. 2 dated February 25, 2022).“

Point 3: Please provide more detail on how the 19 parameters were measured (platforms, in-house vs reference labs, etc).

Response 3: Process was described in Materials and Methods Section: “Venous blood served as a material for the evaluation. It was taken in the clinic’s procedure rooms in properly color-coded vacuum blood collection tubes and prepared in the laboratory in compliance with national standarts. All parameters were measured Sysmex KX-11 and Sapphire 400 Premium platforms at licensed in-house laboratory. The laboratory undergo an audit  of Federal system of external assessment of the quality of clinical laboratory research for every 3 months. Internal laboratory research’s quality audit carried out daily according with Ministry of Health’s orders and National Standart GOST 53133-2008.”

Point 4: Were there changes in weight/BMI after treatment?

Response 4: Weight/BMI changes analysis was made and information was added to the Results section: “Weight and body mass index of patients were analyzed. Patients' weight at arrival was in range of 54-150 kg, mean value is 83 kg. Body mass index (BMI) was in range 20.02-38.63 and mean value is 27.13. Patients' weight at departure was in range of 51-145 kg, mean value is 80 kg. Body mass index (BMI) was in range 19.2-37.23 and mean value is 26.86. Nobody increased his or her weight. Only one patient got BMI 19.2 considered as weight deficiency at departure. The group of those who decrease biological age has mean BMI value 28.68 (in range of 20.08-28.63) and mean value of weight drop is 4 kg. The group of patients who increase biological age has mean BMI value 23.74 (in range of 20.02-29.07) and mean value of weight drop is 2 kg. Pearson’s linear correlation coefficient between weight (BMI) and changes of biological age is -0.47 and -0.5 respectively.”

Also Discussion section got sentence “Linear correlation between weight/BMI changes and biological age changes is near -0.5 proving some connection of weight loss efficiency with metabolism improvement and changes of aging processes.”

Point 5: It is very important to plot the changes for subjects for each parameter and use of simple statistics. It is important to see these parameters and the statistical changes within and not just purely rely upon principal component analysis.

Response 5: Biological age changes were presented on a new Figure 1 considering real patient’s age in Results section.

Point 6: Please provide more specifics on drugs (route, source, dose, frequency, etc.) that the subjects were given.

Response 6: Full specification of the drugs courses are presented in the new Table 1 materials. Brief explanation was added to the Methods section: “All mentioned drugs are orally given herbal food supplements or teas except gastrointestinal adsorbent Enterosgel registered as a medication. The entire information about their  composition and courses are given in Table 1. All patients were examined by Clinic’s physicians to exclude possible contraindications for such therapy. All patients were daily observed by a physicians during the treatment.”

Point 7: Analysis of sex-specific effects should also be presented.

Response 7: All patients were divided into groups according to their age, sex and biological age changes. The results are presented in Table 3 added to the article. Difference of values changes was also presented. I add this text to the Discussion section: “All patients were divided into groups according to their gender and age in compliance with WHO classification. The results of the analysis is presented in Table 3. Comparing two major groups of age 25-44 and 45-60 we observed an almost equal proportion of those who decrease biological age (71 and 68%) but it is 2-3 times more efficient in 45-60 age group. In those who increase their biological age we observed 2 times more significant rise of biological age in 45-60 age group also. All patients who increased values of their biological age are women. Because of all the women should visit a gynecologist in the Clinic (and 20 of 23 did it) their gynecological anamnesis was analyzed. None of the sex-specific trends (menstrual cycle phase, menopause, hormone therapy) was found to be connected with changes.”

Point 8:  How do the biological age changes from this intervention compare to age changes from other interventions.

Response 8: There are a lack of researches about changes of biological age due to interventions on humans in free access. We found no study estimating biological age changes in a short period such as 2 weeks to compare our results. There are articles considering correlation between biological and real age of people in general population without serious ilnesses but not the changes of the first one upon any intervantionThis study is the first research about biological age in medical spa sphere and we hope it can inspire other researches to estimate any interventions with biological age.

Point 9: The author should reiterate the limitations of the study (no control group, the error rate for the model (5.9 years)) in the discussion section.

Response 9: The limitation of the absence of a contorl group is revealed at Response 1. The error rate situation was reevaluated and described at Response 10

Point 10: “The capability of 2-week complex medical spa programs, including calorie restriction diet, medical spa (physiotherapy for gallbladder cleanse and enterosorption, stimulation of the excretory organs), herbal medicine, hydrotherapy, and thermal procedures, to reduce human biological age (Aging.AI 3.0) has been confirmed.” The mean error rate of the model was nearly 6 years. So it is a bit hyperbolic to say biological age was reduced.

Response 10: The mean absolute model error evaluates mean deviation of estimated biological age from real chronological age for this model. But it does not determine a model or its results are incorrect because biological age of persons can not equals their chronological age and should not be close to it neither. We have evaluated only biological age difference between two calculations of it and have not done statements about difference between chronological and biological age involving the model error. These two groups have statistically significant difference according to their p values. So with all respect we conclude the programs can reduce human biological age.

Also the distribution of results is not chaotic. The trend for more significant biological age changes in older patient groups is observed and only women increase biological age values. The underneath biological reasons for this can be supposed.

This explanation was added to the end of Discussion section.

Reviewer 2 Report

The paper “Blood Markers of Biological Age Evaluates Clinic Complex Medical Spa Programs” by Isaev et al reports the effect of a special medical spa program on the blood markers of biological age of patients.  

The study is quite interesting although I have some observation:

  • in material and methods I think that the sample has to be better described (e.i. female/male ratio, subjects/age distribution etc);

  • in results I suggest to better describe figure 1 indicating the meaning of PC1 and PC2

  • always in results table 1 requires, in my opinion, a better explanation; 

  • I suggest to move figure 2 in the results paragraph and add a better explanation beside a increment in quality;

  • In conclusion, a possible application of the spa methods may sound a good way to demonstrate how to use the obtained results to improve the quality of aging.

Overall, considering that the used parameters are strictly connected with age and sex, I think that it could be useful to know the age of the subjects in whom the change was observed because the parameters considered may have different physiological values in different age groups (24-61 is a too wide range). The same thing, as mentioned by the authors, applies to sex: I suggest doing the analysis by dividing the sample into males and females.  

Author Response

Point 1: in material and methods I think that the sample has to be better described (e.i. female/male ratio, subjects/age distribution etc);

Response 1: Gender group analyzis was performed during work at point 6 so link to the corresponding table was written in Materials section. Information about smokers was added also.

Point 2: in results I suggest to better describe figure 1 indicating the meaning of PC1 and PC2

Response 2: The additional figure and corresponding comments was added into the Results secton:

“This PCA biplot (Figure 1) is a combination of loading plot and PCA plot. Nineteen vectors corresponds with each laboratory parameter’s contribution to both of the principal components. Values of their proections to every principal component show parameter’s impact on it. The angle between the vectors reveals the level of correlation between the parameters: 0 degrees means positive correlation, 90 degrees means zero correlation and 180 degrees is negative correlation. Every plot point distinguish a patient’s laboratory profile (red points for arrival profiles and green for the departure ones). The biplot proves t

The second principal component (PC2) mainly contributes to the distribution into arrival and departure groups.”

“Local discriminant Fisher analysis is a statistical method aims to distinguish two or more object classes better. Patient’s laboratory profiles at arrival and departure are such groups. Every plot point is one lab profile (red points are arrival profiles and green are departure profiles). Color area is a zone where a blood markers profile of corresponding group locates with the maximal probability. We see the areas slightly overlap each other assuming significant difference of the profiles between two groups.”

Point 3: always in results table 1 requires, in my opinion, a better explanation;

Response 3: This table was renamed as Table 2 after the article correction. The text was corrected “Table 2 shows a numeric description of the loading plot (Figure 1) and the strength of contribution for each parameter in the principal component calculation.”

Point 4: I suggest to move figure 2 in the results paragraph and add a better explanation beside a increment in quality;

Response 4: This figure was placed to the Results section and explained: “Spearman’s correlation matrix displays correlation coefficients between lab analytes inside a profile and  biological age. The circle size indicates absolute values of correlation coefficients. Colors show negative for orange or positive for blue direction of a correlation. We found biological age correlates with cholesterol, HDL and glucose mostly. They are the same analytes what brings the most contribution to primary component 2 formation.”

Point 5: In conclusion, a possible application of the spa methods may sound a good way to demonstrate how to use the obtained results to improve the quality of aging

Response 5: The conclusion was improved with this exact statement: “Medical spa programs demonstrate a possibility to improve the quality of aging”

Point 6: I think that it could be useful to know the age of the subjects in whom the change was observed because the parameters considered may have different physiological values in different age groups (24-61 is a too wide range). The same thing, as mentioned by the authors, applies to

sex: I suggest doing the analysis by dividing the sample into males and females

Response 6: All patients were divided into groups according to their age, sex and biological age changes. The results are presented in Table 3 added to the article. Difference of values changes was also presented. I add this text to the Discussion section: “All patients were divided into groups according to their gender and age in compliance with WHO classification. The results of the analysis is presented in Table 3. Comparing two major groups of age 25-44 and 45-60 we observed an almost equal proportion of those who decrease biological age (71 and 68%) but it is 2-3 times more efficient in 45-60 age group. In those who increase their biological age we observed 2 times more significant rise of biological age in 45-60 age group also. All patients who increased values of their biological age are women. Because of all the women should visit a gynecologist in the Clinic (and 20 of 23 did it) their gynecological anamnesis was analyzed. None of the sex-specific trends (menstrual cycle phase, menopause, hormone therapy) was found to be connected with changes.”

Reviewer 3 Report

The authors describe the effect on biological age of a 14-day spa program. The brief report is interesting for scientific purpose however methods must be augments. Protocol must be described in a scientific manner. Calorie restriction must be quantified. Protocols of the SPA various treatments must be explained qualitatively and quantitatively. "Drugs" used must be addressed. As it is now it is very hard, to say the least to evaluate the findings. Clearly the results and the conclusion will need adjustments according to the changes made. 

Author Response

Point 1: Protocol must be described in a scientific manner

Response 1: Protocal was described and detailed information are given in Response 2-4 accordingly.

Point 2: . Calorie restriction must be quantified.

Response 2: Calorie restriction and diet principles were described at Materials and Methods section: “All patients underwent a low-calorie gluten-free diet corresponded with the food combining diet principles under observation of a dietologist. Food for the first three days consisted of vegetable and fruit decoctions (200-250 ccal per day). Then the diet was gradually expanded with mild food (puree soups, fruit mousses, vegetable salads) from 400 up to 1200 ccal per day for the next four days. Second week diet brings proteins back to the ration such as eggs, fish and poultry meat with 1600-1700 ccal per day restriction.”

Point 3: Protocols of the SPA various treatments must be explained qualitatively and quantitatively

Response 3:  The program protocol was described in the text in Materails section: “The program consist of medical and spa procedures according to a standard protocol. Medical procedures are: 3 stimulation of biliary excretion procedures (tubages), 3 instrumental colon cleansing procedures with 7 enemas, 12 therapeutic back massages, course of 5 cryotherapy sessions and 7 systemic magnetotherapy sessions. Spa procedures are: wrapping (4 wet and 2 salt wraps) 12 hydrotherapy procedures (jacuzzi, bathtubs, showers at patient's choice), thermal procedures (2 finnish saunas with body peeling and daily available infrared sauna), 13 fitness sessions”

Point 4: "Drugs" used must be addressed.

Response 4: Full specification of the drugs courses are presented in the new Table 1 materials. Brief explanation was added to the Methods section: “All mentioned drugs are orally given herbal food supplements or teas except gastrointestinal adsorbent Enterosgel registered as a medication. The entire information about their composition and courses are given in Table 1. All patients were examined by Clinic’s physicians to exclude possible contraindications for such therapy. All patients were daily observed by a physicians during the treatment.”

Round 2

Reviewer 1 Report

With the changes made by the authors the manuscript is now suitable for publication.

Reviewer 3 Report

The Authors have made the required changes. The article has been widely revisited according to the requests.